# Towards an Automated Monitoring of RF Activity in Low-Power Wireless Testbeds

Markus Schuß, Carlo A. Boano, and Kay Römer
Institute for Technical Informatics
Graz University of Technology, Austria
E-mail: {markus.schuss, cboano, roemer}@tugraz.at

Jakob Link and Matthias Hollick
Secure Mobile Networking Lab (SEEMOO)
Darmstadt University of Technology, Germany
E-mail: {jlink, mhollick}@seemoo.tu-darmstadt.de

## ABSTRACT

To rigorously benchmark the performance of low-power wireless protocols, it is essential to monitor and quantify the RF activity in a given testing environment. Indeed, unwanted radio interference in the surroundings of wireless nodes may worsen their communication performance. Similarly, an inconsistent RF noise across multiple test runs may prevent the ability to fairly compare their results. Unfortunately, to date, this aspect is often neglected by the community, especially due to the lack of monitoring tools enabling a quantitative assessment of RF activity in large testing facilities. In this paper, we move the first steps towards the creation of a low-cost tool automating the distributed monitoring of RF usage in a low-power wireless testbed. Specifically, we first instrument the latest generation Raspberry Pi devices to sense any ongoing activity on the RF channel, enabling a functionality that is typically not available on off-the-shelf Wi-Fi hardware. We then show that one can synchronize the RF measurements of multiple Raspberry Pi connected to a common Ethernet backbone with an average error below 200 $\mu$s. We further devise exemplary strategies to quantify the difference in RF activity across test runs, and enable the real-time detection of deviations in the current RF channel usage compared to what was measured in earlier runs. We finally showcase the ability to compare the RF activity during several test runs and detect when additional interference was present in the environment, as well as when diverse interference patterns were artificially generated.

**Data availability statement.** The firmware used for the data collection as well as the scripts developed to process the raw data and generate the plots presented in this paper are available at *http://www.iti.tugraz.at/schuss20towards*. The authors commit to keep the data publicly available on this institutional repository for at least three years.

## 1 INTRODUCTION

The research community traditionally validates low-power wireless solutions experimentally on real-world testbeds [20]. A large variety of testbeds exist: from small-scale installations used internally by various research groups [13, 28], to large-scale publicly-available facilities such as FIT-IoTLab [1], Indriya [12], and FlockLab [26].

An aspect that is common to most of these low-power wireless testbeds is that they are located in office or university buildings, i.e., the nodes are deployed in open spaces and dynamic environments. As a consequence, these testbeds are often subject to a level of *uncontrollable RF activity*, e.g., generated by laptops, smart-phones, and other devices used by people operating in close proximity.

This is especially relevant as low-power wireless nodes are highly susceptible to radio interference [6]. For example, the transmissions of IEEE 802.15.4 and Bluetooth Low Energy devices are highly vulnerable to transmissions of surrounding Wi-Fi devices, which share the same frequencies (the 2.4 GHz ISM band), use a wider channel bandwidth (20 to 40 MHz), and operate at a transmission power that is higher by several orders of magnitude (up to 20 dBm).

As a result, when evaluating the performance of low-power wireless protocols and comparing it to the state-of-the-art, it is common to make use of testbeds *during night or during weekends* [6, 10, 16, 21, 22, 39, 42], i.e., when buildings are at their quietest, so to minimize the impact of external interference on the experiments.

However, some occasional RF activity may still be present in the testbed area, e.g., due to the idle activities of Wi-Fi access points installed nearby, or due to night owls working until late. Such RF activity may be sufficient to bias the experiments and lead to wrong conclusions, for example when comparing the reliability of state-of-the-art protocols, which is nowadays often close to 100% [7, 14].

Generalizing, *whenever benchmarking protocol performance*, it is important to account for the inherent variability of the experimental conditions and to detect any deviations in the RF environment. The same holds true when carrying out experiments involving the generation of artificial radio interference to stress-test protocols (e.g., using tools similar to JamLab-NG [37]): the synthetic interference patterns should remain consistent throughout different runs and no uncontrolled RF noise should be present in the surroundings. Only this way, one can ensure *reproducible and comparable results*.

Ideally, such an RF activity monitoring is fully automated and integrated into the experimentation chain, i.e., offered by low-power wireless testbed facilities, as highlighted by Boano et al. in an open manifesto to the community [5]. This way, the testing infrastructure can autonomously refute and rerun measurements: for example, in case the RF activity largely varies from that of previous runs.

**Challenges.** However, in order to integrate such functionality in existing testing facilities, several challenges need to be tackled.

*Accurate monitoring of RF activity on a large scale.* First, one needs the ability to observe the RF spectrum across an entire testbed installation. One approach to do this consists in using low-power wireless nodes (e.g., TelosB nodes and nRF52840 dongles) spread across the testbed to scan the received signal strength.

However, besides introducing extra costs, this approach is not optimal due to the limited channel bandwidth of these devices, which makes them unsuitable to accurately detect Wi-Fi activity. Using Wi-Fi devices to fulfil the same task is not feasible, as Wi-Fi hardware does not allow developers to measure RF activity. Therefore, one currently has to resort to spectrum analyzers and software defined radios [3, 23], which is very expensive and does not scale when testbed installations span several floors or large buildings.

*Synchronization of distributed RF measurements.* A second challenge is to establish a common timebase in order to correlate and fuse the RF measurements of several nodes. However, depending on the employed hardware, this may be complex: for example, when connecting RF monitoring devices via USB, the jitter of the FTDI interface makes it hard to accurately time-stamp their measurements [35].

*Quantitative assessment of RF activity during a run.* The ability to monitor the RF spectrum, alone, is insufficient. Without a *metric* quantifying the RF activity during a test run, indeed, only a visual inspection of the RF channel is possible, which is subjective and only allows a *qualitative* assessment [25, 38]. Instead, to rigorously benchmark protocols and claim reproducibility and comparability, a *quantitative* assessment is necessary. To this end, one needs to identify which data a device should collect to objectively and unambiguously quantify the amount of RF activity in its surroundings. Moreover, when using this data to derive a metric capturing the amount of RF activity, one should be able to filter the transmissions of low-power wireless nodes that are part of the testing facility (i.e., the devices running the solution being tested). Without doing so, the computed metric would not only capture the amount of RF noise, but also the "spectrum friendliness" of the tested solution. Ideally, one would have the ability to distinguish between the two.

*Comparing RF activity across test runs.* Finally, as the ultimate goal is to compare whether different test runs have been executed under similar settings, one needs to weigh the currently-measured RF activity against that of previous runs. Specifically, it should be possible to juxtapose the metrics computed across different runs and return whether there were major deviations in RF activity (e.g., additional RF noise or different interference patterns). One should not only account for *temporal* deviations, but also for *spatial* discrepancies, as wireless nodes are typically spread across a large area. This comparison process should ideally require a limited amount of time and not be resource-intensive. This way, right at the end of an experiment, one can deem whether a rerun is necessary.

**Contributions.** In this paper, we tackle these challenges and move the first steps towards the creation of a low-cost tool automating the monitoring of RF activity in low-power wireless testbeds.

We first show that it is possible to use the Wi-Fi module embedded on off-the-shelf Raspberry Pi 3B+/4 hardware to monitor RF usage at sufficient granularity to recognize common interference sources in the 2.4 GHz band. This is important, as these devices are often already used as observer nodes in low-power wireless testbeds to orchestrate activities, measure performance, and generate RF noise [8, 29, 35, 37]. We achieve this by using Nexmon, a C-based firmware patching framework for Cypress Wi-Fi chips [33, 34].

We then show that one can synchronize the RF measurements of multiple Raspberry Pi 4B (RPi4) connected to a common Ethernet backbone (i.e., in the same way as observer nodes are connected in

a testbed facility), with an average error below 200 $\mu$s. This enables us to correlate distributed RF measurements and devise exemplary strategies to quantify the difference in RF activity across test runs.

Specifically, we use the distribution of the observed power over time at the various nodes and illustrate different techniques allowing the real-time detection of deviations in the RF channel usage compared to what was measured in earlier runs. We further show how this approach allows to filter the activity of a testbed's own nodes and showcase the ability to detect when additional radio interference was present in the environment, as well as when diverse interference patterns were artificially generated by JamLab-NG.

After describing related work in § 2, this paper proceeds as follows:
- We instrument RPi4 devices to monitor nearby RF activity (§ 3).
- We illustrate how we can observe the same RF activity across multiple RPi4 with low synchronization errors (§ 4).
- We describe how to quantify the difference and detect deviations in the measured RF activity across several test runs (§ 5).
- We close the paper in § 6 along with a discussion on future work.

## 2 RELATED WORK

To account for variations in RF activity and avoid inconsistencies across test runs, researchers often monitor the RF spectrum and determine if its usage is steady. To this end, they use low-cost spectrum analyzers such as the Wi-Spy[1] [23, 25, 31, 38], or low-power wireless nodes to sample the received signal strength [15, 18]. However, this process mostly consists in a visual inspection of the RF channel usage, which is subjective and only allows a qualitative assessment. Instead, we aim to provide a quantitative assessment of RF activity.

A few low-power wireless testbed facilities (e.g., FlockLab [41], TWIST [11, 23, 24], and w-iLab.2 [40]) embed software-defined radios, Wi-Spy, or high-end spectrum analyzers to allow their users to monitor RF activity in the surroundings of the testbed nodes. However, they have just one monitoring node across the testbed [3], operate in sub-GHz frequency only [41], or rely on old Wi-Fi hardware with limited functionality, such as the ath9k chips [40]. Moreover, they do not perform any synchronization of the distributed measurements and do not endeavour to compute a metric *quantifying* the RF activity, so to enable a better reproducibility and comparability of results, which is the ultimate goal of our work.

A few researchers have analyzed the RF activity on a channel using off-the-shelf hardware for different purposes. Hermans et al. [19] aim to identify the source of interference in IEEE 802.15.4 networks. Similarly, Grimaldi et al. [17] aim to classify external interference in real-time via supervised learning. Noda et al. [30] try to quantify the quality of the channel to build interference-aware wireless sensor networks. Brown et al. [9] measure the probability distribution function of idle periods to estimate the packet reception rate of an IEEE 802.15.4 network before deployment. These works monitor the RF channel with the goal of *mitigating* radio interference. In contrast, in this work, we aim to perform a distributed RF monitoring *to account for the inherent variability* of the conditions in the testing environment and inform the user accordingly.

Puccinelli et al. [32] have proposed a metric capturing key properties of the network topology that may affect protocol performance. Such metric allows to recognize whether performance variations

---

[1]https://www.metageek.com/products/hardware/

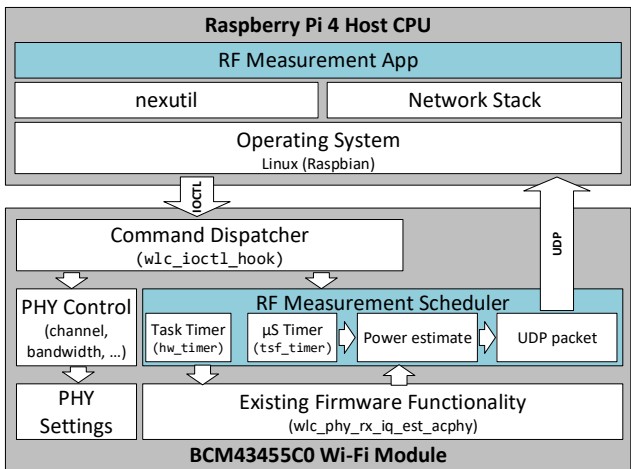

**Figure 1: Sketch of our RF monitoring functionality on RPi4. A userland application (RF Measurement App) interacts with a scheduler running within the BCM43455C0 radio firmware (RF Measurement Scheduler) in order to collect a sequence of RF power estimates.**

across experiments are due to properties of the network topology. Our aim is close in spirit, but *with focus on RF activity*, i.e., we ultimately aim to provide a mechanism assessing whether performance variations across runs are caused by changes in RF activity.

## 3 MONITORING SURROUNDING RF ACTIVITY USING OFF-THE-SHELF WI-FI HARDWARE

Our aim is to design a *low-cost* solution to monitor the RF activity within a low-power wireless testbed. To this end, as discussed in §2, the use of specialized hardware such as spectrum analyzers and software-defined radios (SDR) is not an option due to their high costs. Indeed, even the cheapest SDR costs over 100 €, and requires a dedicated powerful computer to orchestrate its operations. For this reason, instrumenting a testbed with several of these high-end devices would be both expensive and labor-intensive.

Also using a fraction of the low-power wireless nodes embedded in the testbed is not an option to directly observe the RF spectrum. Indeed, tools such as TI's SmartRF Studio[2], Nordic Semiconductors' nRF Connect RSSI viewer[3], and Contiki's RSSI scanner[4], have only a limited channel bandwidth: one would either require several nodes to monitor a single Wi-Fi channel, or let a single node continuously shift frequency at a cost of a lower sampling rate. Moreover, these low-power radios would need to be connected to one of the observer nodes in the testbed for further data processing or storage.

The ideal case would be to *reuse a testbed's observer nodes* for this purpose. For example, many low-power wireless testbeds make use of devices such as the Raspberry Pi as observer nodes to orchestrate activities, measure performance, and generate RF noise [27, 29, 35, 37]. As these devices embed radio modules operating in the 2.4 GHz

band, they could be used to also monitor the surrounding RF activities. Unfortunately, the Raspberry Pi 3B and later revisions embed a Wi-Fi module, but they do not allow to measure the strength of the RF signal at an arbitrary point in time – a problem that is common to most off-the-shelf Wi-Fi hardware[5].

In this work, we tackle this limitation and *enable off-the-shelf observer nodes to monitor the surrounding RF activity*. While also the Cypress (former Broadcom) BCM43455C0 Wi-Fi module found on recent Raspberry Pi devices does not provide a way to instantaneously measure the strength of the RF signal, one can use reverse engineering and flash patching tools to craft an RF power estimator on these low-cost Wi-Fi modules. To this end, we use Nexmon, a C-based firmware patching framework that has been used in the past to i.a., enable monitor mode on Cypress Wi-Fi chips [34].

Thanks to Nexmon, one can already record each individual Wi-Fi transmission (even those from networks with which a device is not associated) using tools such as Wireshark or tcpdump, and derive a list of sniffed packets in pcap format. However, besides Wi-Fi activity, no other source of RF noise can be currently monitored.

Therefore, we extend Nexmon as follows. First, we make use of Ghidra[6], a software reverse engineering suite, to gleam into the inner workings of the BCM43455C0 firmware and spot leftover functionality that is usually not accessible to end-users (e.g., hidden functions that are only partially implemented, as well as a remnant of calibration and compliance testing features). We identify one function (wlc_phy_rx_iq_est_acphy), that fits exactly our purposes: it instructs the RF front-end to compute the power estimate over a given number of samples ($2^{10}$ in our implementation). While the power estimate returned by this function is sufficient to monitor RF activity (we use this value in the remainder of this paper), a manual calibration is needed to express the power estimate in dBm.

Building upon this function, we create a userland application (RF Measurement App) and a scheduler running within the BCM43455C0 radio firmware (RF Measurement Scheduler) that interact in order to collect a sequence of RF power estimates, as shown in Fig. 1. Specifically, we use Nexmon's nexutil tool to trigger commands for the RF measurement scheduler using input/output control (IOCTL) system calls. As the overhead of the system calls is significant, polling the radio for RF power measurements would result in a limited and non-deterministic sampling rate. Therefore, similar to the approach used in JamLab-NG [37], we make use of the IOCTL interface to only instruct the radio to begin periodic measurements on a specific channel with a given bandwidth.

Internally, the RF measurement scheduler uses a timer (hw_timer) to periodically call the wlc_phy_rx_iq_est_acphy function. We timestamp the power estimates returned by this function with a μs-precision timer (tsf_timer), and generate a UDP packet to be injected into the wlan0 interface. Each UDP packet contains a single timestamped power estimate and it is sent to port 5555, such that it can be captured by the RF measurement app accordingly.

---

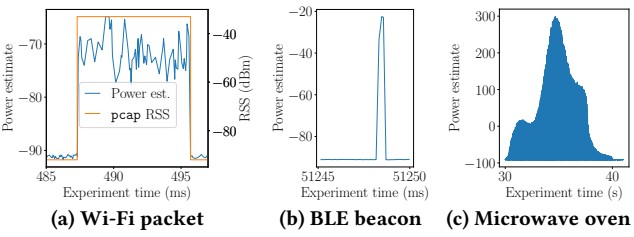

**(a) Wi-Fi packet**   **(b) BLE beacon**   **(c) Microwave oven**

**Figure 2: Power estimates returned by a `RPi4` in the presence of different devices generating RF noise in the 2.4 GHz band.**

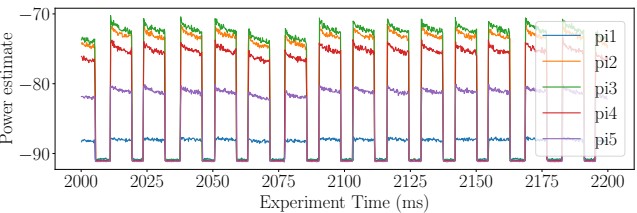

**Figure 3: Power estimate returned by five `RPi4`, spread over 16 $m^2$ in a room, observing the same source of Wi-Fi traffic.**

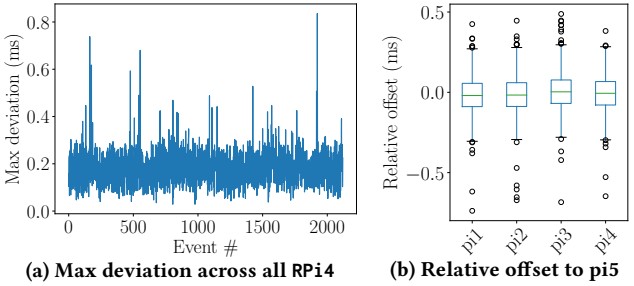

**(a) Max deviation across all RPi4**   **(b) Relative offset to pi5**

**Figure 4: Synchronization error across five `RPi4` monitoring the same RF activity. The boxes and whiskers in Fig. 4b show the median (green center line), the first and third quantile (box body), as well as the 1.5 interquartile range (whiskers).**

Following this procedure, we can instrument the `RPi4` to estimate surrounding RF activity at 7.8 kHz[7]. Fig. 2 shows exemplary power estimates returned by the RF measurement app in the presence of Wi-Fi and BLE traffic, as well as a microwave oven operating nearby. The Wi-Fi traffic is generated by a second `RPi4` placed 2 m away, whereas the BLE traffic is generated by an nRF52840DK node placed 30 cm away. The microwave oven was used to heat up water and was located 1.5 m away from the `RPi4` sniffing nearby RF activity. All measurements make use of a channel bandwidth of 20 MHz.

We also place a third `RPi4` using Nexmon's monitor mode and `tcpdump` to capture the duration and strength of the generated Wi-Fi traffic in a `pcap` file. This third `RPi4` is also placed 2 m away from the one generating Wi-Fi traffic and its measurements are synchronized with those of the `RPi4` sniffing RF activity as described in § 4. Fig. 2a shows that the power estimate returned by the sniffing `RPi4` (blue line) correctly captures the over-the-air duration of the Wi-Fi packet extracted by `tcpdump`'s `pcap` file (orange line).

# 4 SYNCHRONIZATION OF DISTRIBUTED RF MEASUREMENTS

To measure the RF activity across large-scale testbed installations (i.e., to have a good *spatial* coverage) and to monitor the activities in the entire 2.4 GHz ISM band (i.e., to monitor several Wi-Fi channels at once), several `RPi4` devices should be used to perform RF measurements at the same time. Therefore, it is important to synchronize their activities and establish a common time-base. Given that the time between two samples is 128 µs, the timesync error between any two `RPi4` should ideally be in the same range or lower.

To this end, we have implemented time-stamping twice throughout the measurement chain shown in Fig. 1. We first timestamp the collected power estimates using the RF measurement app. To do this, we employ the `RPi4`'s `unix` timestamp: as this is the operating system's time, it can be kept in sync across different `RPi4` in a testbed using the network time protocol (NTP), as shown in [37]. With all devices attached to the same wired network, the NTP implementation used (*chrony*), reported an estimated offset of >30 ns and a standard deviation of >20 µs. These numbers depend on the size and topology of the network, but should not exceed 100 µs for common testbeds using Ethernet to connect the `RPi4` devices.

However, as each power estimate is independently passed from the RF front-end to the RF measurement app through the operating

system and its network stack, one experiences non-deterministic delays affecting the accuracy of individual samples.

Therefore, we add a second timestamp as soon as the power estimates have been sampled by the RF front-end, i.e., right after the `wlc_phy_rx_iq_est_acphy` function has returned, using the time synchronization function timer (`tsf_timer`[8]). This second timestamp provides a more fine-grained resolution that allows to accurately account for ephemeral RF events (e.g., a sequence of short BLE beacons). Hence, one could use this timestamp to correct the `unix` timestamps added by the RF measurement app.

Following this procedure, we instrument five `RPi4` located in the same room and interconnected by an Ethernet backbone to sense the ongoing RF activity on the same Wi-Fi channel. We also place in the same room another `RPi4` running JamLab-NG to generate periodic Wi-Fi packets. Fig. 3 shows the power estimates collected from each `RPi4` using the `unix` timestamp. As expected, due to the different location of the nodes and their distance from the `RPi4` generating Wi-Fi traffic, the absolute value of the estimated power is different. However, each spike, which corresponds to the on-air time of a Wi-Fi packet, is well-synced across the five `RPi4`.

To better quantify the synchronization error across the different `RPi4`, we consider more than 2000 Wi-Fi packets and compare the timestamp of each rising edge in the estimated power (i.e., the beginning of each spike in Fig. 3). Fig. 4 shows the maximum deviation across all the five `RPi4` and the relative error to a specific device (pi5). Regardless of which `RPi4` is used as reference, the median synchronization error including all uncertainties in our

---

[7]Note that one can achieve a higher rate by decreasing the number of samples and by sending several measurements in a single UDP packet. In this work, we focus on a prototypic implementation and leave these optimizations as future work.

[8]The `tsf_timer` is used by Nexmon's monitor mode to perform time-stamping at the MAC layer and keep synchronized the Wi-Fi stations connected to the same access point (AP). However, as a `RPi4` is not connected to an AP when collecting power estimates, its measurements are not automatically synced to those of nearby nodes.

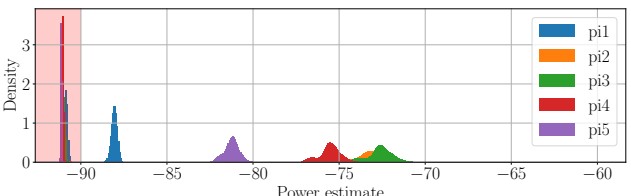

**Figure 5: Probability density function of power estimates computed with a 300 s window on 5 RPi4 observing the same Wi-Fi activity. The red portion marks noise floor samples.**

measurement chain does not deviate by more than 22 µs, with 95% of the samples never exceeding an error of 209 µs.

As shown in Fig. 3 and 4, the synchronization accuracy of the `unix` timestamp is quite satisfactory. One can further increase accuracy by correcting these timestamps using those obtained with the `tsf_timer`. Moreover, as the `unix` timestamps are already drift-corrected using NTP, one could use linear regression to calculate a correction factor for the timestamps obtained with the `tsf_timer`. The latter exhibits a drift that strongly depends on the temperature of the `RPi4`, which varies as a function of its computational load[9].

## 5  QUANTIFYING THE DIFFERENCE IN RF ACTIVITY ACROSS TEST RUNS

With the ability to measure the RF activity using off-the-shelf `RPi4` outlined in § 3 and § 4, one can manually inspect the measurements and look for outliers. As this results in subjective and qualitative assessments only (see § 2), in this section we derive a metric that allows to determine in real-time whether the ongoing RF activities are similar to those recorded in a previous experiment. Note that our aim is not to derive the *best* metric to compare the RF conditions in different experiments, but rather to showcase the feasibility of such real-time comparison as a seed for future work in the area.

**Selecting a metric.** In order to enable a real-time detection of deviations in the RF channel usage compared to what was measured in earlier runs, a first necessary step is the selection of a metric capturing the large number of power estimates sampled over time into a compact representation. While one could model or learn different RF interference patterns and compare those with the currently measured power estimates, we choose to make no assumptions about the characteristics of the RF activity and forgo any training. Instead, we derive a probability density function (PDF) of the observed power estimates over a time window. This approach is more generic, it allows to account for the strength of the RF signal (e.g., to capture whether sources of interference have moved closer or further away over time), and is more practical than learning in the presence of *several* sources of RF noise at the same time.

Fig. 5 shows the PDF computed for five `RPi4` deployed in the same configuration used earlier (i.e., in the presence of an additional `RPi4` in the same room generating periodic Wi-Fi traffic) over a time window of approximately five minutes. The region marked in red indicates the absence of RF noise (i.e., the noise floor of each `RPi4`). Although the PDF shown in Fig. 5 allows to capture the ongoing RF activity measured by each `RPi4`, it does not contain a fine-grained

___________
[9]We observed clock differences in the range of 5 ms within 10 minutes (i.e., 8 *ppm*).

information about the power estimates in the time domain. For this reason, RF interference occurring for a short amount of time gets averaged out and cannot be accounted for. To mitigate this problem, one can simply shorten the observation window, such that one can also account for ephemeral RF interference.

**Quantifying deviations in RF channel usage.** In order to enable an automatic comparison of RF activity between runs, we investigate how to quantitatively compare two PDFs (such as the one shown in Fig. 5). To this end, we reuse existing methods included in the popular computer vision suite `opencv` to compare histograms[10]. Among others, we make use of correlation (Eq. 1), Hellinger distance (Eq. 2), as well as Kullback-Leibler divergence (Eq. 3). The first two are bounded between $[0, 1]$ but behave differently for small discrepancies; the latter is an example of an open-ended scale. These three methods are defined as follows:

$$d(H_1, H_2) = \frac{\sum_I (H_1(I) - \bar{H}_1)(H_2(I) - \bar{H}_2)}{\sqrt{\sum_I (H_1(I) - \bar{H}_1)^2 \sum_I (H_2(I) - \bar{H}_2)^2}} \quad (1)$$

$$d(H_1, H_2) = \sqrt{1 - \frac{1}{\sqrt{\bar{H}_1 \bar{H}_2 N^2}} \sum_I \sqrt{H_1(I) \cdot H_2(I)}} \quad (2)$$

$$d(H_1, H_2) = \sum_I H_1(I) log\left(\frac{H_1(I)}{H_1(I)}\right) \quad (3)$$

where $N$ is the number of histogram bins, $H_k(I)$ represents the bin of histogram $k$ using a power estimate $I$, $\bar{H}_k = \frac{1}{N} \sum_J H_k(J)$, and $d(H_1, H_2)$ is a representation of the "distance" across histograms computed based on the three aforementioned techniques.

We compare the deviation in RF activity across different runs using these three methods as follows. Using the same setup illustrated previously, we let five `RPi4` record power estimates on Wi-Fi channel 7 (2442 MHz) over 5-minutes runs. During each run, a nearby Raspberry Pi 3B using JamLab-NG generates a reproducible interference pattern on the same Wi-Fi channel during the first minute and the last three minutes (i.e., no interference is generated from 60 s to 120 s). We also collocate two TelosB nodes running Contiki in the same room. These two nodes periodically exchange 8 packets/sec using `nullmac`, `nullrdc`, and `Rime` on IEEE 802.15.4 channel 18 (2440 MHz); logging the packet reception rate (PRR), i.e., the number of correctly received packets over time.

Fig. 7 shows the recorded power estimates of an exemplary `RPi4`, as well as the PRR of the TelosB nodes for three different runs. Whilst the first and the second run are identical, in the third run we purposely create changes in the RF environment: a short burst of strong interference at 90 s lasting five seconds and a switch to a different (lighter) interference pattern in the last two minutes of the runs, i.e., after 180 s. Note that the spikes in Fig. 7a at 120 s and 240 s are due to a periodic self-calibration function of the Wi-Fi module and can be easily filtered due to their high value (up to 5000).

Fig. 6 shows the deviation between the three runs using the aforementioned histogram comparison methods. To generate the histograms for the comparison, we employ a time window of 1 s and compute 128 bins in the range $[-92, 0]$. While correlation has the advantage of having an upper bound, obtained by the comparison of

___________
[10]https://docs.opencv.org/3.4/d6/dc7/group__imgproc__hist.html#ga994f53817d621e2e4228fc646342d386

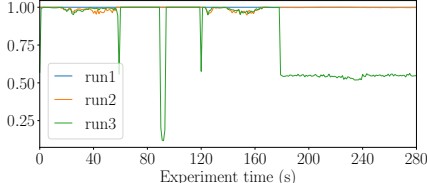

(a) Correlation

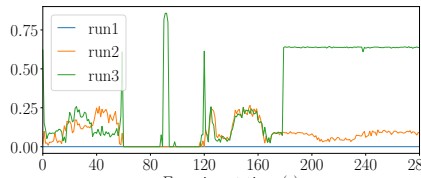

(b) Hellinger distance

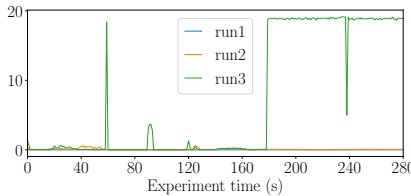

(c) Kullback-Leibler divergence

**Figure 6: Deviation between the power estimates obtained in three exemplary runs (see Fig. 7a) using three different methods.**

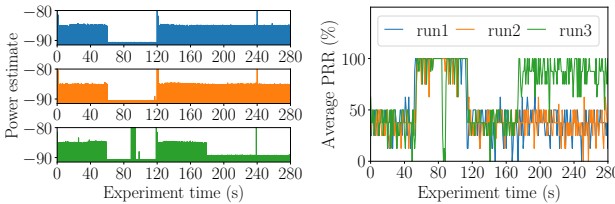

(a) Recorded power estimate     (b) PRR of TelosB nodes

**Figure 7: Recorded power estimate by a `RPi4` (a) and PRR of two TelosB nodes (b) in the presence of Wi-Fi interference.**

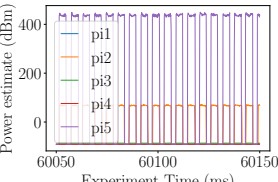

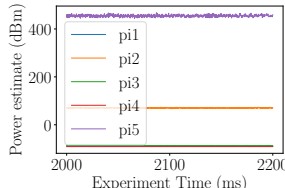

(a) IEEE 802.15.4 packets     (b) Modulated carrier tone

**Figure 8: Power estimate for five `RPi4` spread out across a room observing a single TelosB generating RF activity.**

the first run (run1) with itself, the Hellinger distance and Kullback-Leibler divergence start from 0 for identical histograms and increase proportionally to the difference between runs. From these results, we can conclude that the Hellinger distance is especially sensitive to small changes, whereas correlation does not penalize smaller deviations in the RF activity. Conversely, the Kullback-Leibler divergence can capture and significantly penalize ephemeral changes such as the self-calibration spike at 240 s.

All three methods clearly identify the artificial changes in RF usage introduced in the third run and are suitable to detect significant deviations in RF activity. To ultimately assess whether one should invalidate a test run, one can use Fig. 7b to gauge the impact of the changes in RF activity on the PRR between the TelosB nodes. Note that all three methods are computed within a fraction of a second while the computation of the histogram takes roughly 2 s on a single core of a modern processor. Hence, one can quickly detect deviations in the RF channel usage at the end of a test run, and autonomously make a decision on whether refuting the results and re-running the same experiment, as envisioned in [5].

**Filtering the activity of a testbed's own nodes.** So far, we have only focused on the detection of surrounding RF activity and ignored the impact of the transmissions of co-located low-power wireless nodes. However, when integrating such a solution into a low-power wireless testbed facility, one should be able to filter the transmissions of low-power wireless nodes that are part of the testing facility (i.e., the devices running the solution being tested).

Traditionally, such low-power wireless nodes are directly attached to the observer nodes in the testbed, i.e., they are located in very close proximity. Our experiments have actually shown that one can easily recognize and filter transmissions of low-power wireless nodes located in very close proximity to the `RPi4` due to the high magnitude of the power estimate. Fig. 8 shows the power estimate of five `RPi4` devices in the presence of a TelosB node sending packets periodically (a) and emitting a continuous modulated carrier tone (b) as in [4] using a transmission power of 0 dBm. The TelosB node is attached to pi5 and it is located about 50 cm away from pi2, 1 m away from pi3, and 2 m away from pi1 and pi4. As Fig. 8 shows, the `BCM43455C0` heavily overestimates the narrowband signal to over 400 on pi5, whereas pi2 still reports a power estimate of about 50, which is easily distinguished from surrounding RF interference, which typically returns a lower power estimate value. At about 2 m the signal is indistinguishable from the noise floor, and cannot be detected by pi1 and pi4. Therefore, one can, in principle be agnostic to the transmissions of the nodes attached to a `RPi4` acting as observer node in a low-power wireless testbed.

## 6 CONCLUSIONS AND FUTURE WORK

When benchmarking the performance of low-power wireless systems, it is important to account for the inherent variability of the RF conditions in the testing environment. In this paper, we have put the basis for the creation of a low-cost tool automating the distributed monitoring of RF activity in low-power wireless testbeds. After instrumenting several Raspberry Pi 4B nodes to monitor the RF activity in their surroundings and synchronizing their measurements, we have showcased the ability to quantitatively compare the RF usage during several test runs and detect critical deviations.

Our work represents an important step towards a better reproducibility and comparability of results. However, to ensure that the experimental conditions are *exactly* the same across multiple runs, it is not sufficient to only check the amount of RF activity in the surroundings of the wireless nodes. For example, the ability to monitor that the link quality between the nodes in the testbed (which may vary if nodes are slightly moved, nearby shelves are moved, and doors are opened) did not change across multiple runs, is an orthogonal effort that goes beyond this paper. Similarly, the exemplary strategies to quantify the difference in RF activity across test runs presented in this paper only allow to objectively conclude how similar the RF conditions were when running several experiments: determining whether the variability of the RF conditions is sufficient to deem two or more test runs as comparable is not in the scope of this work. In the future, we plan to tackle also these issues, and to integrate our full-fledged RF monitoring approach into the framework of an existing benchmarking facility (e.g., D-Cube [36]).

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
