# OpenReview forum: "Towards an Automated Monitoring of RF Activity in Low-Power Wireless Testbeds"
_sigmobile.org/MobiCom/2020/Workshop/CPS-IoTBench — CPS-IoTBench 2020_

### Official Review · AnonReviewer1 · 2020-06-25
**very interesting paper that presents a novel technique to measure RF activity**

**Rating:** 7
**Confidence:** 3

**Review:**

In this paper, the authors design and prototype a tool to monitor the RF activity
in low-power wireless testbeds. Testbeds usually suffer from significant and highly
variable interference patterns. Running repeatable and comparable experiments in such
testbeds is difficult without a thorough quantification of the underlying RF noise. To measure
and quantify this, the authors propose a cheap solution using off-the-shelf raspberry pi
devices with a modified wireless card firmware. The solution is then validated using
real-world experiments using multiple raspberry pi nodes.

Strengths:
1. Reproducibility and fair experiment comparisons are major issues in science. Working
toward achieving this in wireless testbeds is laudable and should be considered an important topic.
2. Low-cost solutions such as using off-the-shelf RPIs and just modifying their software
is a very attractive approach.
3. The authors have a convincing set of experiments.

Questions:
1. Scalability: the authors have used what seems like a limited testbed (just a few
meters between devices etc.). It remains unclear how this solution would work in larger
testbeds and how accurate it would be.
2. Completeness/Soundness: It is unclear why and how the PDF approach (in sec 5) is a good
discriminant between two RF activities. It would be interesting if the authors could motivate
this in more depth because one could probably craft two different interference patterns
that sum up to the same PDF. However, it is likely that they influence the experiments
in different ways.

---

### Official Review · AnonReviewer2 · 2020-06-26
**Clear paper, opens door for discussion**

**Rating:** 9
**Confidence:** 5

**Review:**

This paper outlines a RPi based distributed architecture for monitoring RF activity in the area of a low power testbed. It motivates the use of RPi as many testbeds already have them, then outlines how to use the on-board WiFi to do the monitoring. One of the more complex elements, which is well explained, is the issue of time synchronizing the multiple RPis so that the RF observations can be correlated.

One point that is not clear to me is regarding the channel observations. As the RPis sniff on WiFi, they are actually looking at a spectrum which is much wider that of the 802.15.4 nodes. Is there any negative (or positive?) effect to this mismatch? Also, are some channels better adapted to this monitoring than others, e.g., does this work best when the 802.15.4 channel is "in the middle" of the wifi frequency?

It was noted that one motivation for using the RPi to make the measurements is that they are already in use by the testbeds. Nevertheless, precise timing is required, and much of the paper is dedicated to what must be done on the RPi to get the accurate timing. Is there any potential interference between these measuring/timing properties and the other activities that the RPi might be performing to support the testbed?

Regarding the presented metric: the example provided does a good job to outline how it measures "similarity", but I wonder how applicabile this metric is to a wide range of protocols and/or interference patterns. it certainly offers a concrete option, but the door remains option to other metrics.

Is it correct that all measurements of interference presented in the paper come from jamlab generated interference? If so, are the authors planning additional experiments with non-emulated wifi?

While the above could be perceived as negative points, I believe that instead they can provide points of discussion at the workshop, and that the paper will provide a valuable contribution for discussion.

---

### Official Review · AnonReviewer4 · 2020-06-26
**interesting paper with very valuable contribution**

**Rating:** 9
**Confidence:** 4

**Review:**

The authors present a framework for measuring the background RF activity. The measured background RF activity can be then used to decide whether several runs  of the benchmarks are comparable and whether the experiment needs to be repeated or not. This framework will provide researchers with valuable information whether two experiments were executed under similar conditions.

To achieve this goal, the researchers built a test-bed utilising several Raspberry Pies, which makes the test-bed affordable and easy to build.

The evaluation is thorough and a lot of focus is given to the problem of synchronisation. This problem was well addressed and the authors properly describe and evaluate their solution.

Because the proposed solution is based on Raspberry Pi to sniff the traffic the Wi-Fi is used. Because the WiFi channels are much wider (i.e. 22 MHz vs 2 MHz in the case of 802.15.4), how would the framework work if the sensor node transmits at different 802.15.4 channels within the larger Wi-Fi channel. Also, will their solution be able to distinguish between two nodes transmitting on two different 802.15.4 channels within the same 802.11 channel?

Overall, I found the paper very well written and easy to follow. The contributions of the paper are sound and well evaluated. I highly recommend this paper to be included into the workshop proceedings.

---

### Official Review · AnonReviewer3 · 2020-07-01
**Interesting paper that is exciting, but over-promises and under-delivers; should prompt valuable discussions though**

**Rating:** 6
**Confidence:** 4

**Review:**

Overall, this is an interesting paper and a good incremental step forward in tackling a very challenging problem.

## Major Contributions

 - *Accessible* spectrum monitoring solution. Significant enhancement of state-of-art monitoring capability effectively without requiring additional / custom RF hardware.
 - Discussion opener on techniques for compactly describing channel conditions in a manner appropriate to low-power, wireless protocol designers.

## Major Drawbacks

 - **The second-to-last sentence of §5 almost makes the whole paper feel deceitful.** The whole premise of this work as introduced is to detect interference for low-power protocol design. The introduction goes on to explain that testbeds are in real-world environments, full of possibly interfering devices. The experiments present nice demonstrations of detecting interference, multiple devices, quantifying, etc. Then at the very end, it's finally revealed that this system **cannot detect interference from other low-power devices in the environment?**
    - This is made (much) worse by Fig 2(b), which makes it look like BLE will be detected by this system. Upon further reading, however, it seems that signal is only seen because the BLE "interferer" is 30cm away in that experiment. In real-world scenarios, the ambient Bluetooth devices that will interfere with protocols under study will not all be within 30cm (or perhaps within 2m given Fig 8) of the measurement station.
 - Much is written about time synchronization, but nothing answers "how good does it need to be to be useful?"
 - The proposed "metric" feels fairly arbitrary, but this is probably acceptable given the context of the rest of the paper.

----

## Detailed Feedback for Authors

The introductory / abstract framing is a bit too harsh; I don't think it's fair to say that 'inconsistent noise' is 'largely neglected by the community' -- there is a reason that wireless protocol results require long-duration studies and why things like the EWSN dependability competition take multiple runs to compute a score. It's not neglected, it's simply that the community does not (yet) have accessible tools to measure this, so the best solution available is to take many samples over a long period to mitigate the impact of (hopefully random) noise as much as possible [indeed, the giant list of cites in the second column bolsters that view; re-framing this as a contribution towards addressing a known problem would be both less antagonistic and more accurate].

What is the significance of 200µs synchronization? The argument for synchronization for correlating readings from multiple sample points of the testbed is compelling, but it needs to include a discussion of _how good the sync must be to be useful_. Especially given the later discussion about NTP time versus BCM time.

Data availability: Two years isn't very compelling for a testbed artifact, as testbeds tend to last decade+. Consider something like Zenodo (which has a ~20yr guarantee at the moment) as well?

In related, "or rely on old Wi-Fi hardware.." -- what is bad about old Wi-Fi hardware? Old radios still send and receive signals don't they? [I think footnote 5 on the next page answers this, but this idea needs to pull up into the main text in related. Being old is not inherently a reason to dismiss something].

For future work: I'm curious about the calibration -- is that stable / fixed per device or will it drift with time / environmental conditions.

For time sync:
 - How well is NTP expected to work? How much does this rely on network topology?
    - This is certainly well-studied / known, but pulling in a sentence would help the reader understand the limitations of this sync approach
 - "One can hence use this timestamp to correct the `unix` timestamps added by the RF measurement app" -- how? Specifically, the intro suggests issues surrounding syscall latency and other interference / jitter sources. Are these being addressed in this sync design? If so, how?
 - The results of 22µs and 209µs would be much more compelling if there was a baseline of expectations presented first
 - "The synchronization accuracy of the `unix` timestamp **is quite satisfactory**" -- for what? No goal was ever given for timestamp accuracy, which makes it hard to understand from this evaluation why this is "satisfactory".

For the metric:
 - Future work suggestion: There are several channel models that include well-defined interference. It would be valuable to apply the metric to these deterministic interference scenarios to give a baseline for the meaning of the metric.
 - Please refrain from introducing unnecessary variables (θ); "window" is a fine term.
 - I did not expect to have to choose between capturing short-term interference or long-term trends. For future efforts, some effort to combine these phenomena would be valuable
 - The metric exploration is not terribly compelling. It reads a bit like a survey of available openCV functions run at random. This is somewhat understandable given that this is a very new space / idea, but this could be greatly improved by discussing _why_ each of the histogram comparison mechanisms are chosen, what kind of interference they are expected to detect, and whether it worked.

---

#### One small thing

I can't believe I'm already *that reviewer*, but I had some technology struggles reading this paper. Several of the figures brought iPad + Notability to a *crawl* (especially Fig 7a, which froze for nearly a minute trying to render). Please try to downsample traces like these a bit where there is far more data than can realistically be viewed anyway.

---

# Summary

Overall, I think this will bring valuable discussions to the workshop. I do wish that this workshop had a 'conditional acceptance' mechanism, as I would gate this on being far more upfront about the limitations of the interference detection. That alone dropped my score from an easy accept to just on the line.

---

### Decision · Program_Chairs · 2020-07-07

Accept